# Congenital diaphragmatic hernia in a middle-income country: Persistent high lethality during a 12-year period

Ana Sílvia Scavacini Marinonio[1]*, Milton Harumi Miyoshi[1], Daniela Testoni Costa-Nobre[1], Adriana Sanudo[1], Kelsy Catherina Nema Areco[1], Mandira Daripa Kawakami[1], Rita de Cassia Xavier Balda[1], Tulio Konstantyner[1], Paulo Bandiera-Paiva[1], Rosa Maria Vieira de Freitas[2], Lilian Cristina Correia Morais[2], Mônica La Porte Teixeira[2], Bernadette Cunha Waldvogel[2], Carlos Roberto Veiga Kiffer[1], Maria Fernanda Branco de Almeida[1], Ruth Guinsburg[1]

1 Escola Paulista de Medicina – Universidade Federal de São Paulo (UNIFESP), São Paulo, São Paulo, Brazil, 2 Fundação Sistema Estadual de Análise de Dados (SEADE Foundation), São Paulo, São Paulo, Brazil

* anascavacini@gmail.com

**Data Availability Statement:** The database of the study was uploaded and available in the ZENODO

## Abstract

### Background

In high- and middle-income countries, mortality associated to congenital diaphragmatic hernia (CDH) is high and variable. In Brazil, data is scarce regarding the prevalence, mortality, and lethality of CDH. This study aimed to analyze, in São Paulo state of Brazil, the temporal trends of prevalence, neonatal mortality and lethality of CDH and identify the time to CDH-associated neonatal death.

### Methods

Population-based study of all live births with gestational age ≥ 22 weeks, birthweight ≥400g, from mothers residing in São Paulo State, Brazil, during 2004–2015. CDH definition and its subgroups classification were based on ICD-10 codes reported in the death and/or live birth certificates. CDH-associated neonatal death was defined as death up to 27 days after birth of infants with CDH. CDH prevalence, neonatal mortality and lethality were calculated and their annual percent change (APC) with 95% confidence intervals (95%CI) was analyzed by Prais-Winsten. Kaplan-Meier estimator identified the time after birth that CDH-associated neonatal death occurred.

### Results

CDH prevalence was 1.67 per 10,000 live births, with a significant increase throughout the period (APC 2.55; 95%CI 1.30 to 3.83). CDH neonatal mortality also increased over the time (APC 2.09; 95%CI 0.27 to 3.94), while the lethality was 78.78% and remained stationary. For isolated CDH, CDH associated to non-chromosomal anomalies and CDH associated to chromosomal anomalies the lethality was, respectively, 72.25%, 91.06% and

public repository. The DOI for the data is: 10.5281/zenodo.6992692.

**Funding:** This research was supported by Fundação de Amparo à Pesquisa do Estado de São Paulo (FAPESP), Project # 2017/03748-7. FAPESP had no role in study design, data collection and analysis, decision to publish, or preparation of the manuscript.

**Competing interests:** The authors have declared that no competing interests exist.

97.96%, during the study period. For CDH as a whole and for all subgroups, 50% of deaths occurred within the first day after birth.

## Conclusions

During a 12-year period in São Paulo State, Brazil, CDH prevalence and neonatal mortality showed a significant increase, while lethality remained stable, yet very high, compared to rates reported in high income countries.

## Introduction

Congenital Diaphragmatic Hernia (CDH) has an estimated global prevalence of 2.3:10,000 live births [1], ranging from 1.09:10,000 live births in Iran between 2011 and 2016 to 3.2:10,000 live births in Florida between 1998 and 2012 [2, 3]. In Florida, CDH prevalence among live births was stationary between 1998 and 2012 [3], as well as in Barbados, another high-income country, between 1993 and 2012 [4].

CDH phenotypic presentation is variable, being reported as isolated CDH in more than 70% of cases or associated to non-chromosomal anomalies in around 22% and to chromosomal anomalies in 3–4% [3, 5]. Several variables are related to survival of patients with CDH, such as the phenotypic presentation, the lung size, the presence of liver herniation and pulmonary hypertension, the pregnancy interruption practices, procedures performed in utero, the clinical management during transition from fetal to neonatal life, and the quality of perinatal care [6–8].

Lethality of patients with CDH is high [7], especially during the neonatal period, reaching up to 30% in high-income countries [8]. For Latin American Centers, isolated CDH lethality during the neonatal period is 68%, varying from 23% for mild cases to 97% for severe CDH [6]. Considering CDH associated to congenital anomalies, the lethality rate reported by one middle-income country referral center was 64% [9].

In Brazil, some single centers studies reported lethality rates in infants with CDH up to 27 days after birth between 27% and 89% [5, 10], reaching 100% for CDH associated to chromosomic anomalies [10]. One population-based study done in São Paulo, with data retrieved from live births and mortality information system without linkage, available at DATASUS (Informatic Department of the Brazilian Health System), showed 64% of lethality rate between 2006–2017 [11]. However, there are no studies that address the temporal trend of CDH prevalence, mortality, and lethality in the Brazilian states or in the country that can provide consistent information to guide public health policies for CDH management.

Therefore, the objectives of this study were to analyze, in São Paulo state of Brazil, the temporal trends of prevalence, neonatal mortality and lethality of CDH and its subgroups (isolated CDH, CDH associated to chromosomal anomaly and CDH associated to non-chromosomal anomalies) and, finally, to identify the time to CDH-associated neonatal death.

## Methods

This is a population-based study of all live births with gestational age $\geq$22 weeks and birthweight $\geq$400g from mothers residing in São Paulo State, Brazil, between 2004 and 2015. São Paulo State, with around 41 million inhabitants, is responsible for 32% of the national income, with the second highest Human Development Index in the country (HDI 0,783 in 2010) [12] and is listed as the 21st biggest economies of the world [13].

In the State, all information regarding births and deaths are filled in by physicians in the Live Births and Deaths certificates, respectively. The physician should follow a manual prepared by the Brazilian Ministry of Health [14, 15], in which the notification and codification of congenital anomalies is mandatory [16]. After completion, these certificates are forwarded to the Health Offices and Registry Offices [14, 15], which send monthly electronic records of the information on the Live Birth and Infant Death Certificates (up to 365 days after birth) to the SEADE Foundation. Based on live birth and death information, SEADE Foundation makes the deterministic linkage from death to live birth records in order to identify birth information from all infants who died within 365 days after birth [17]. The data set was organized according to the study design towards a final database integrating common birth variables among the neonatal death records and live birth records, ensuring 99% of success in linkage between both datasets [18].

The study was approved by the Ethics Committee of Universidade Federal de São Paulo, under the number 4.055.489, with waived informed consent.

CDH definition was based on ICD-10 codes found in any line of the death and/or live birth certificates and classified into 3 subgroups: isolated, CDH associated to chromosomal anomalies and CDH associated to non-chromosomal anomalies (Table 1).

Non-chromosomal anomalies associated to CDH were further classified into nervous system (Q00-Q07), eye, ear, face, and neck (Q10-Q18), circulatory system (Q20-Q28), respiratory system (Q30-Q34), cleft lip and cleft palate (Q35-Q37), digestive system (Q38-Q45), genital organs (Q50-Q56), urinary system (Q60-Q64), musculoskeletal system (Q65-Q79), and other congenital anomalies (Q80-Q89). CDH-associated neonatal death was defined as any death of infants with CDH (according to Table 1) that occurred between 0 and 27 completed days after birth.

The following demographic characteristics were described for all CDH live births and CDH-associated neonatal deaths: maternal age (<20, 20–34 and ≥35 years), maternal schooling (<12 and ≥12 years), parity (primiparous or multiparous), number of prenatal care visits (0, 1–6 and ≥7), pregnancy (single or multiple), delivery mode (vaginal or cesarean section), gestational age (preterm or term), birthweight (<2500g or ≥2500g), sex (male or female), 1st and 5th minute Apgar score (0–6 and ≥7).

Prevalence, neonatal mortality, and lethality rates of CDH and its subgroups (isolated CDH, CDH associated to non-chromosomal anomalies and CDH associated to chromosomal

**Table 1. CDH definitions.**

| Classification | ICD-10 codes |
|---|---|
| CDH | Q79.0—congenital diaphragmatic hernia |
| | K44—diaphragmatic hernia |
| | K44.0—diaphragmatic hernia with obstruction, without gangrene |
| | K44.1—diaphragmatic hernia with gangrene |
| | K44.9—diaphragmatic hernia without obstruction or gangrene |
| Isolated CDH | CDH codes with/or without the following: |
| | Q33.6—pulmonary hypoplasia |
| | Q33.3—pulmonary agenesis |
| | Q25.0—patent ductus arteriosus |
| CDH associated to chromosomal anomaly | CDH codes AND any code between |
| | Q90 and Q99 (Chromosomal abnormalities) |
| CDH associated to non-chromosomal anomaly | CDH codes AND any other chapter Q ICD-10 code except those used in the definition of isolated CDH or CDH associated to chromosomic anomaly) |

anomaly) were calculated. Prevalence was calculated by the number of live births with CDH per 10,000 live births; neonatal mortality was obtained by the number of CDH-associated neonatal deaths per 10,000 live births; and lethality was calculated by the number of CDH-associated neonatal deaths per 100 live births with CDH. Prevalence, neonatal mortality, and lethality annual trends were analyzed by Prais-Winsten Model, and their annual percent change (APC) with 95% confidence intervals (95%CI) were calculated.

Kaplan-Meier estimator was applied to identify the time from birth to the CDH-associated neonatal death for each CDH group during the 12-year study period.

All procedures were done using Stata 15.1® (StataCorp LLC, Texas, USA).

## Results

From 2004 to 2015, there were 7,317,611 live births in São Paulo State, Brazil. Among them, 7,314,257 were included in the study. CDH was reported in 1,225 (0.02%) live births, with 965 (78.8%) CDH-associated neonatal deaths (Fig 1).

More than one anomaly group was found in 128 (35.8%) of the 358 live births with CDH associated to non-chromosomal anomalies. Anomalies in the musculoskeletal system were the most frequently group associated to CDH (S1 Table). Among the chromosomal anomalies associated to CDH, Trisomy 18 was the most frequent (S1 Table).

Demographic characteristics of CDH live births and CDH-associated neonatal deaths are described in Table 2.

### Prevalence of CDH

CDH prevalence, during 12-year study period, was 1.67 (95%CI 1.60 to 1.77) per ten thousand live births. For isolated CDH, CDH associated to non-chromosomal anomalies, and CDH associated to chromosomal anomalies, the prevalence was, respectively, 1.12, 0.49 and 0.06 per ten thousand live births (S2 Table).

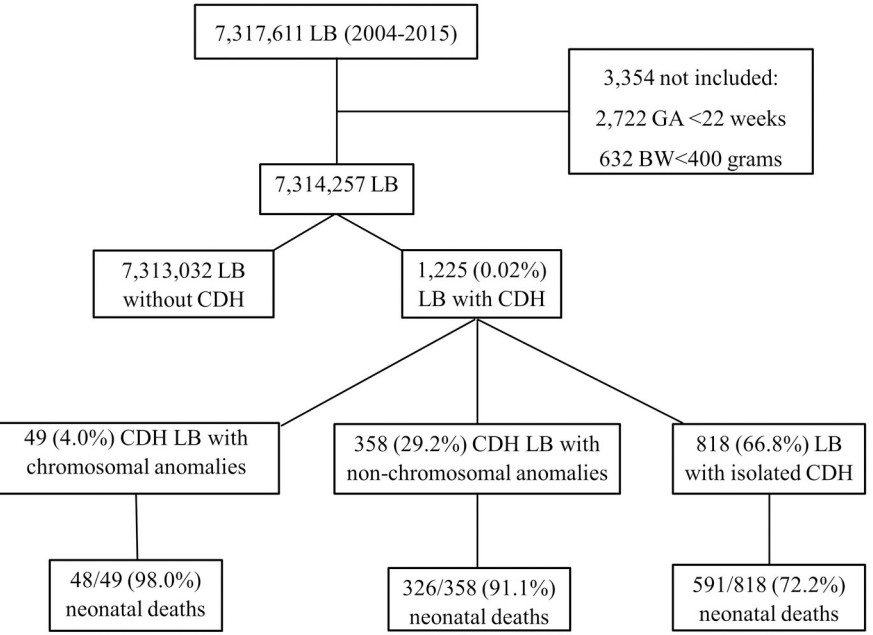

**Fig 1. Flowchart of included infants.** LB: live births; CDH: congenital hernia diaphragmatic; GA: gestational age; BW: birthweight.

**Table 2. General characteristics of CDH live births and CDH-associated neonatal deaths.**

|  | CDH live births | CDH-associated neonatal deaths |
|---|---|---|
| *Maternal age (years)* | n = 1,225 | n = 965 |
| <20 | 13% | 14% |
| 20–34 | 68% | 68% |
| ≥ 35 | 19% | 19% |
| *Maternal schooling (years)* | n = 959 | n = 751 |
| ≤ 7 | 26% | 28% |
| 8–11 | 54% | 55% |
| ≥ 12 | 20% | 17% |
| *Primiparous* | n = 997 | n = 795 |
| Yes | 43% | 41% |
| *Prenatal care visits* | n = 1,195 | n = 940 |
| 0 | 1% | 1% |
| 1–6 | 30% | 31% |
| ≥ 7 | 69% | 67% |
| *Pregnancy* | n = 1,225 | n = 965 |
| Single | 97% | 97% |
| *Delivery* | n = 1,225 | n = 965 |
| Cesarean section | 72% | 72% |
| *Gestational age (weeks)* | n = 1,204 | n = 952 |
| 22–27 | 2% | 3% |
| 28–31 | 6% | 7% |
| 32–36 | 28% | 28% |
| 37–41 | 64% | 62% |
| ≥ 42 | 1% | 1% |
| *Birthweight (grams)* | n = 1,220 | n = 963 |
| Mean (range) | 2577 (500–4750) | 2518 (500–4610) |
| ≥ 2500 | 61% | 57% |
| *Sex* | n = 1,224 | n = 965 |
| Male | 57% | 55% |
| *1st minute Apgar score* | n = 952 | n = 741 |
| ≥ 7 | 25% | 17% |
| *5th minute Apgar score* | n = 954 | n = 743 |
| ≥ 7 | 57% | 48% |

CDH: congenital diaphragmatic hernia; n: total number available for the variable.

The annual trend of CDH prevalence rate showed an increasing pattern, with a positive Annual Percent Change (APC): 2.55; 95%CI 1.30 to 3.83. An increase in annual rates was also observed for isolated CDH (APC 2.11; 95%CI 0.54 to 3.70), and for CDH associated to non-chromosomal anomalies (APC 3.96; 95%CI 0.87 to 7.15). For CDH associated to chromosomal anomalies, the annual trend was stationary (APC 1.85; 95%CI -7,82 to 12,53) (Fig 2).

## CDH neonatal mortality and lethality

CDH neonatal mortality rate, was 1.32 per ten thousand live births, with an increasing annual trend (APC 2.09; 95%CI 0.27 to 3.94). The lethality rate during the study period was 78.78%, with a stationary pattern (APC -0.43; 95%CI -1.83 to 0.10) (Fig 3).

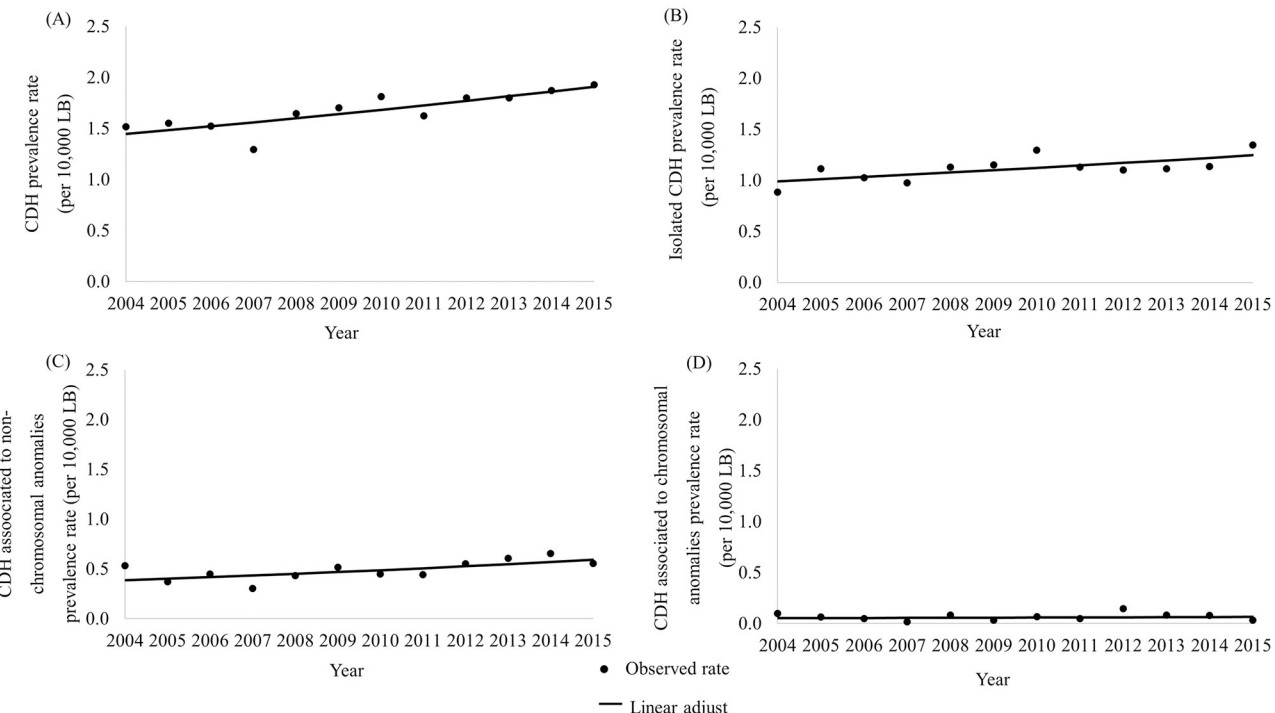

**Fig 2. Annual trend of CDH prevalence rate adjusted by Prais-Winsten model.** CDH prevalence rate per year adjusted by Prais-Winsten analysis; (A) Any CDH; (B) Isolated CDH; (C) CDH associated to non-chromosomal anomalies; (D) CDH associated to chromosomal anomalies. CDH: congenital diaphragmatic hernia; LB: live births.

The neonatal mortality and lethality data, as well as their annual trends, for CDH subgroups are shown in Table 3.

Kaplan-Meier analysis showed that the median time of CDH-associated neonatal deaths was 24 hours after birth (95%CI 24 to 25 hours). For isolated CDH the median time of deaths was 26 hours (95%CI 24 to 31 hours); for CDH associated to non-chromosomal anomalies the median time to death was 24 hours (95%CI 16 to 24 hours), and for CDH associated to chromosomal anomalies, 23 hours (95%CI 7 to 48 hours) (S1 Fig).

## Discussion

This populational study evaluated the temporal trends of CDH prevalence, neonatal mortality, and lethality during a 12-year period in São Paulo State, Brazil. CDH prevalence was 1.67 per 10,000 live births, with a significant increase throughout the period. CDH neonatal mortality also increased throughout the period, while the lethality remained stable between 2004–2015. Considering all CDH subgroups, isolated CDH represented the majority of cases, and its prevalence increased over the years. No change in neonatal mortality rate temporal trends was identified among CDH subgroups. Lethality for CDH and for all subgroups were high, and most CDH-associated neonatal deaths occurred within the first day after birth.

In the present study, CDH prevalence was lower than 2.1 per 10,000 live births reported by some high and middle-income countries between 2004 and 2015 [19], nevertheless similar to that found in São Paulo State between 2006 and 2017 [11]. However, contrasting to a stationary prevalence reported by high income countries [3, 4], our study showed that CDH prevalence increased over the years. This finding is similar to those observed for all congenital anomalies in Brazil and may be the result of case notification increase due to training and

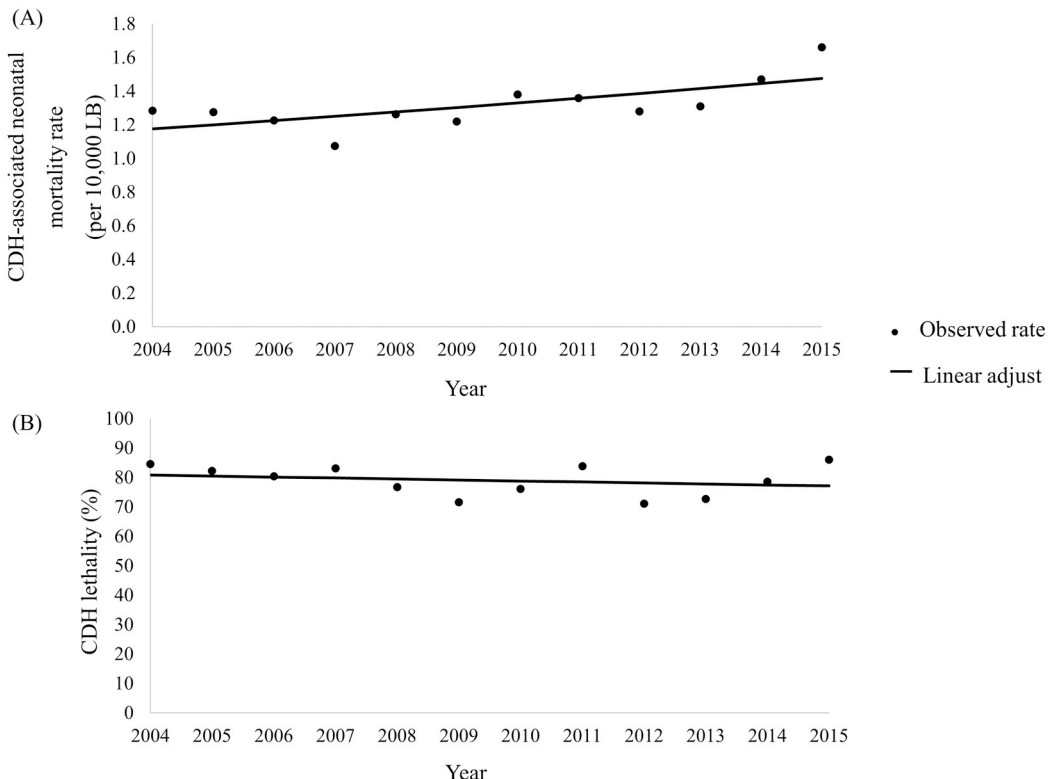

**Fig 3. Annual trend of CDH neonatal mortality rate and CDH lethality adjusted by Prais-Winsten model.** (A) CDH neonatal mortality rate per year adjusted by Prais-Winsten analysis; (B) CDH lethality per year adjusted by Prais-Winsten analysis. CDH: congenital diaphragmatic hernia; LB: live births.

monitoring for congenital anomalies report, promoted by the Brazilian Ministry of Health [16]. In addition, improved access to the health care system [20] and better perinatal emergency care, with a greater chance of survival beyond the delivery room [21], may have contributed to improvements in CDH identification.

The increased prevalence and the fact that CDH lethality remained high during the study period may have resulted in the CDH neonatal mortality increase observed over the years in São Paulo State. The lethality of CDH reported in our study is much higher than the 20.2% found in high-income countries [3], or the 33.5% found in a multi country population-based study [8]. Considering that CDH is a rare congenital malformation that requires complex and multidisciplinary care, antenatal diagnosis is important for maternal referral to tertiary and surgical centers prior to delivery [22, 23], where experts are available, as well as protocols for initial stabilization in the delivery room, postnatal care, and optimal clinical and surgical treatment options [22]. Based on these considerations, the high CDH neonatal mortality and lethality observed in our study may reflect São Paulo State health inequities [24]. São Paulo State has discrepancies in access to primary and qualified health care [24], a lack of adequate strategic planning to intensive care development [25], inequities in resources distribution and care effectiveness [26]. All these factors, associated with the lack of evidence-based literature to guide neonatal care in terms of ventilatory, hemodynamic and sedation for the CDH management [27], possibly contribute with the high neonatal mortality rate and lethality observed in São Paulo State during the study years [28].

**Table 3. Neonatal mortality rate, lethality and annual trends for Congenital Diaphragmatic Hernia subgroups.**

|  | Isolated CDH | CDH associated to non-chromosomal anomalies | CDH associated to chromosomal anomalies |
|---|---|---|---|
| **Neonatal mortality** |  |  |  |
| • Per 10,000 live births | 0.81 | 0.45 | 0.07 |
| • Annual trend | stationary | stationary | stationary |
| • Annual Percent Change—% (95% confidence interval) | 1.98 (-0,45 to 4.46) | 2.38 (-0.71 to 5.58) | 1.85 (-7.82 to 12.53) |
| **Lethality** |  |  |  |
| • Per 100 live births with CDH (by CDH subgroup) | 72.25 | 91.06 | 97.96 |
| • Annual trend | stationary | decreasing | * |
| • Annual percent change—% (95% confidence interval) | -0.096 (-1.889 to 1.730) | -1.57 (-2.34 to -0.79) | * |

CDH: congenital diaphragmatic hernia;

*Prais-Winsten model do not fit, and it was not possible to calculate trends.

Furthermore, comparing our CDH lethality with the 65% reported by another population-based study carried out in the state of São Paulo using DATASUS [11], possibly the greatest consistency of our database, with more than 99% of linkage between neonatal death records and live birth records, resulted in better retrieval of information and the higher lethality found in our study [18, 29].

CDH-associated neonatal deaths occurred earlier in São Paulo State, compared to South American data reported by ECLAMC, which showed 0.2% of deaths on the first day after birth [8]. Probably most CDH-associated neonatal deaths in São Paulo State occurred during the first day after birth due to the lack of strategies for initial stabilization and inadequate infrastructure to implement cardiopulmonary support, critical to enhance survival chances of newborn infants with CDH [7, 30].

In terms of lethality, our study showed higher rates than the Latin American CDH Study Group, which reported 68.1% of lethality for isolated left CDH [6]. In our study, 553/818 (67.6%) of isolated CDH were reported on the live birth certificates, but it is possible for the rate of antenatal diagnosis to be much lower. The isolated CDH subgroup is the one which benefits the most from antenatal diagnosis [31] and prenatal procedures, thus improving the prognosis of neonates with moderate pulmonary hypoplasia [32].

Associated non-chromosomal anomalies considerably reduce survival rates in neonates with specific congenital anomalies [3]. The fact that the lethality of this CDH subgroup decreased over the study period deserves further analysis in order to identify these survivors. Despite this trend, lethality was exceddingly high in this subgroup, over 90% throughout the period.

Finally, in the present study, the subgroup of infants with CDH associated to chromosomal anomalies represented 4% of total CDH infants, with Trisomy 13 and 18 accounting for 67% of the cases. European and Brazilian studies also report Trisomy 18 as the most common chromosomal anomaly associated to CDH, followed by Trisomy 13 and 21 [5, 21]. In this subgroup of CDH infants, stationary trends with wide confidence intervals were observed for prevalence and neonatal mortality rates during the study period. The inability to analyze the lethality temporal trend may have been influenced by the number of infants in this subgroup each year. However, in our study, lethality in this group was close to 100%, similar to a report from a Brazilian specialized tertiary center [10], although higher than the 40% observed in a population-

based study in a high-income country [3]. In Brazil, there is no specific guideline for the perinatal management of newborn infants with Trisomy 13 or 18 and many of them do not receive active care, resulting in early deaths [33].

This study has several limitations. The database was provided by SEADE Foundation after linking and anonymization, and this process rely on a manual component which is time-consuming; therefore, the most recent year available for the study was 2015. The database is based on live birth and death certificates; thus, there is a risk of information bias since it depends on the notification of the diagnosis (including the impossibility of confirming the performance of chromosomal tests, which may have influenced the prevalence of chromosomal and non-chromosomal anomalies associated with CDH). The database does not include information on maternal diseases, treatments and procedures during pregnancy, CDH characteristics and management. Despite these limitations, this study is one of the first population-based evaluations of the temporal trends of CDH prevalence and neonatal deaths in a middle-income country and was done using a database that covers a greater volume of data compared to other available databases in Brazil [29], allowing a more reliable understanding of low-occurrence diseases. Considering the economic, social, and health inequities in São Paulo State, further research is necessary to identify CDH prevalence and outcome variation across the state.

In conclusion, during a 12-year period in São Paulo State, Brazil, CDH prevalence and neonatal mortality showed a significant increase, while lethality remained stable, yet high compared to rates reported in high income countries.

## Supporting information

**S1 Fig. Age in hours at the time of CDH-associated neonatal deaths, according to Kaplan-Meier survival estimate.**
(PDF)

**S1 Table. Frequency of chromosomal and non-chromosomal anomalies associated with CDH, per diagnoses.**
(PDF)

**S2 Table. Number of live births, CDH live births and CDH-associated neonatal deaths, per year.**
(PDF)

## Acknowledgments

We thank all technical staff of Fundação Sistema Estadual de Análise de Dados (SEADE Foundation) for their work with the database and Josiane Quintiliano Xavier de Castro, MD, for helping in the deterministic linkage between Live Birth Certificates and Death Certificates.

## Author Contributions

**Conceptualization:** Ana Sílvia Scavacini Marinonio, Milton Harumi Miyoshi, Daniela Testoni Costa-Nobre, Mandira Daripa Kawakami, Rita de Cassia Xavier Balda, Tulio Konstantyner, Mônica La Porte Teixeira, Bernadette Cunha Waldvogel, Carlos Roberto Veiga Kiffer, Maria Fernanda Branco de Almeida, Ruth Guinsburg.

**Data curation:** Kelsy Catherina Nema Areco, Mandira Daripa Kawakami, Paulo Bandiera-Paiva, Rosa Maria Vieira de Freitas, Lilian Cristina Correia Morais, Mônica La Porte Teixeira, Bernadette Cunha Waldvogel.

**Formal analysis:** Ana Sílvia Scavacini Marinonio, Daniela Testoni Costa-Nobre, Adriana Sanudo.

**Funding acquisition:** Maria Fernanda Branco de Almeida, Ruth Guinsburg.

**Investigation:** Ana Sílvia Scavacini Marinonio, Milton Harumi Miyoshi.

**Methodology:** Ana Sílvia Scavacini Marinonio, Milton Harumi Miyoshi, Daniela Testoni Costa-Nobre, Adriana Sanudo, Tulio Konstantyner, Carlos Roberto Veiga Kiffer, Maria Fernanda Branco de Almeida, Ruth Guinsburg.

**Project administration:** Ruth Guinsburg.

**Resources:** Rosa Maria Vieira de Freitas, Lilian Cristina Correia Morais, Mônica La Porte Teixeira, Bernadette Cunha Waldvogel.

**Supervision:** Milton Harumi Miyoshi, Ruth Guinsburg.

**Validation:** Ana Sílvia Scavacini Marinonio, Daniela Testoni Costa-Nobre, Adriana Sanudo.

**Visualization:** Ana Sílvia Scavacini Marinonio, Milton Harumi Miyoshi, Daniela Testoni Costa-Nobre, Maria Fernanda Branco de Almeida, Ruth Guinsburg.

**Writing – original draft:** Ana Sílvia Scavacini Marinonio, Daniela Testoni Costa-Nobre.

**Writing – review & editing:** Ana Sílvia Scavacini Marinonio, Milton Harumi Miyoshi, Daniela Testoni Costa-Nobre, Adriana Sanudo, Kelsy Catherina Nema Areco, Mandira Daripa Kawakami, Rita de Cassia Xavier Balda, Tulio Konstantyner, Paulo Bandiera-Paiva, Rosa Maria Vieira de Freitas, Lilian Cristina Correia Morais, Mônica La Porte Teixeira, Bernadette Cunha Waldvogel, Carlos Roberto Veiga Kiffer, Maria Fernanda Branco de Almeida, Ruth Guinsburg.

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
