## [Decision Letter · Decision Letter 0]

13 Dec 2022

PONE-D-22-22879Congenital diaphragmatic hernia in a middle-income country: persistent high lethality during a 12-year period.PLOS ONE

Dear Dr. SCAVACINI MARINONIO,

Thank you for submitting your manuscript to PLOS ONE. After careful consideration, we feel that it has merit but does not fully meet PLOS ONE’s publication criteria as it currently stands. Therefore, we invite you to submit a revised version of the manuscript that addresses the points raised during the review process.

I am returning your manuscript with two reviews. Note that the manuscript should show more details about the data source.

We look forward to receiving your revised manuscript.

Kind regards,

Xiaohong Li

Academic Editor

PLOS ONE

https://journals.plos.org/plosone/s/fileid=ba62/PLOSOne_formatting_sample_title_authors_affiliations.pdf.

“This research was supported by Fundação de Amparo à Pesquisa do Estado de São Paulo (FAPESP), Project # 2017/03748-7, however with no role in any step of the study and report (including study design, analysis and interpretation of data, in the writing of the report or in the decision to submit the paper for publication), which was authors´ responsibility.”

“We thank all technical staff of Fundação Sistema Estadual de Análise de Dados (SEADE Foundation) for their work with the database and Josiane Quintiliano Xavier de Castro, MD, for helping in the deterministic linkage between Live Birth Certificates and Death Certificates.”

“This research was supported by Fundação de Amparo à Pesquisa do Estado de São Paulo (FAPESP), Project # 2017/03748-7, however with no role in any step of the study and report (including study design, analysis and interpretation of data, in the writing of the report or in the decision to submit the paper for publication), which was authors´ responsibility.”

Additional Editor Comments:

Abstract

1. Line 26 of page 2, I strongly suggest that the incidence of CDH should be modified to birth prevalence of CDH throughout the manuscript. It is very difficult to estimate the incidence of birth defects.

2. Line 42 of page 2, The results showed that the median time of CDH-associated neonatal deaths was 24 hours after birth, so I think the statement 'most deaths occurred within the first day after birth' is not exactly, '50% deaths occurred within the first day after birth' might be more sound.

Introduction

1. Line 51 of page 4, the contents of reference 1 is about prevalence, not incidence.

2. Line 54 of page 4, birth prevalence of CDH is preferred.

3. Line 72 of page 4, please clarify the age group of lethality rate, in neonatal period or in infancy, or others?

Methods

1. I think more details about death certificates and live birth certificates should be given in the methods section, such as who, when and how collect death certificates, as well live birth certificates; the process of birth defect ascertainment; the quality measures of the two dataset.

2. Line 102, The definition and classification of CDH are shown in Table 1. Why are the diagnostic ICD-10 codes, such as K44, classified as CDH? So It's important to clarify the ascertainment time of CDH. Besides, if babies are diagnosed as CDH, will all cases undergo further chromosome examination? This is important to access the trend of prevalence of CDH associated with non-chromosome abnormality.

3. I would like to see the results about survival curve for newborns affected with CDH. That will give the readers more information.

Results

1. Line 138 of Page 8, please check the number '35.8%'. Also, please check all the data shown in Supplementary Table 1. The order of ' Other chromosomal anomalies' and 'Other trisomy' should be inter-changed.

2. Line 147-150 of page 9, could you please show 95%CI of birth prevalence of CDH in the main text.

3. Line 152 of page 9, the abbreviated 'CI' need to be defined earlier.

Reviewers' comments:

Reviewer's Responses to Questions

**Comments to the Author**

1. Is the manuscript technically sound, and do the data support the conclusions?

Reviewer #1: Yes

Reviewer #2: Yes

2. Has the statistical analysis been performed appropriately and rigorously? 

Reviewer #1: Yes

Reviewer #2: Yes

3. Have the authors made all data underlying the findings in their manuscript fully available?

Reviewer #1: Yes

Reviewer #2: Yes

4. Is the manuscript presented in an intelligible fashion and written in standard English?

Reviewer #1: No

Reviewer #2: Yes

5. Review Comments to the Author

Reviewer #1: This is an important contribution to the knowledge on CDH. The authors have presented the data clearly, pointing out the limitations of the data. My once comment relates to the use of the term incidence. Incidence is not used to report the occurrence of congenital anomalies, as the total number of conceptions cannot be measured. Instead, birth prevalence is used. This correction may be made throughout the manuscript.

Reviewer #2: The authors have presented a really nice review of population based data on CDH births, mortality, and lethality in the Sao Paulo State of Brazil. The introduction and methodology of the manuscript are sound. I also thought that the results are presented clearly and succinctly for the limited data available on the topic. The discussion was thorough, but it seemed to go on for a little too long. I would recommend that the authors work to cut some of the discussion wording to make it a quicker and more streamlined read. Otherwise, I think it is worthy of publication and adds to the worldwide education about the status of CDH care in this middle-income country.

6. PLOS authors have the option to publish the peer review history of their article (what does this mean?). If published, this will include your full peer review and any attached files.

Reviewer #1: **Yes: **Anita Kar

Reviewer #2: No

---

## [Author Response · Author response to Decision Letter 0]

3 Jan 2023

Here is a point-by-point response to the editor and reviewers’ comments and concerns.

Comments from Editor

Abstract

• Comment 1. Line 26 of page 2, I strongly suggest that the incidence of CDH should be modified to birth prevalence of CDH throughout the manuscript. It is very difficult to estimate the incidence of birth defects.

Thank you for the comment. We agree with the question and we have changed “incidence” for “prevalence” throughout the text. 

• Comment 2. Line 42 of page 2, The results showed that the median time of CDH-associated neonatal deaths was 24 hours after birth, so I think the statement 'most deaths occurred within the first day after birth' is not exactly, '50% deaths occurred within the first day after birth' might be more sound.

Thank you. We corrected the sentence in the abstract - last line of results (page 2, lines 43-44) as follows: “For CDH as a whole and for all subgroups, 50% of deaths occurred within the first day after birth.”

Introduction

• Comment 1. Line 51 of page 4, the contents of reference 1 is about prevalence, not incidence.

Thank you again for your considerations. Similar to abstract, after your consideration, “incidence” was changed to “prevalence” throughout the text.

• Comment 2. Line 54 of page 4, birth prevalence of CDH is preferred.

Thank you again. The term “incidence” was changed to “prevalence” throughout the text.

• Comment 3. Line 72 of page 4, please clarify the age group of lethality rate, in neonatal period or in infancy, or others?

After the suggestion, the age group was inserted. The text now reads (page 5, lines 71-73) as follows: “In Brazil, some single center studies reported lethality rates in infants with CDH up to 27 days after birth between 27% and 89% [5,10], reaching 100% for CDH associated to chromosomic anomalies [10].”

Methods

• Comment 1. I think more details about death certificates and live birth certificates should be given in the methods section, such as who, when and how collect death certificates, as well live birth certificates; the process of birth defect ascertainment; the quality measures of the two dataset.

Thank you for the questions regarding data collection and quality. The answer is reported in two steps: 1. acquisition and quality of the data and, 2. verification of birth defects.

1. Acquisition and quality of the data: all births and deaths that occur in Brazil are registered through certificates called, respectively, Live Birth and Death Certificates. Completing these certificates is a legal responsibility of the physician, who fills them out in accordance with manuals prepared by the Brazilian Ministry of Health (Brasil, 2022-a; Brasil, 2022-b). Congenital anomalies notification is compulsory, and those detected at birth must be reported in Live Births Certificates, which contain an objective question about the presence or absence of congenital anomalies detected at birth and a field for recording all visible congenital anomalies, which must be described and coded according to the 10th version of the International Statistical Classification of Diseases and Related Problems to Health (ICD-10) (WHO, 2010). Death Certificates are completed after any death and contain information about the cause of death and any diagnoses that contributed to the death, which may include congenital anomalies. Once completed, these declarations are forwarded to the registry offices (Brasil, 2022-a; Brasil, 2022-b). The registry offices of the 645 municipalities in the State of São Paulo send monthly the electronic Live Birth and Infant Death Certificates (up to 365 days after birth) to the SEADE Foundation, which is the institution responsible for collecting, organizing, analyzing and disseminating records of deaths and live births in the State. Based on death and live birth information, SEADE Foundation databases of live births and infant deaths are built using deterministic linkage from death to live birth records in order to identify birth information from all infants who died within 365 days after birth. Identification and creation of true pairs of both certificates is achieved either by equality or similarity, aiming to detect divergences, allowing corrections, and incorporating available information in only one instrument. This procedure is decisive in improving the statistics of the civil registry of the São Paulo State (Waldvogel, 2019).

2. Verification of birth defects: despite mandatory notification, the quality of completion and coding of fields related to congenital anomalies in the Live Birth and Death Certificates may vary among municipalities and health centers. Some anomalies are not visible at delivery and others, although diagnosable at birth, require instruments or specific technical knowledge not always available in all public or supplementary health services, which may result in suboptimal or incorrect records (Nhoncanse, 2012; Brasil, 2021).

After your suggestion, we included, in the methods section (page 6, lines 92-101), a summary of this information, as follow: “In the State, all information regarding births and deaths are filled in by physicians in the Live Births and Deaths Certificates, respectively. The physician should follow a manual prepared by the Brazilian Ministry of Health [14,15], in which the notification and codification of congenital anomalies is mandatory [16]. After completion, these certificates are forwarded to the Health Offices and Registry Offices [14,15], which send monthly electronic records of the information on the Live Birth and Infant Death Certificates (up to 365 days after birth) to the SEADE Foundation. Based on live birth and death information, SEADE Foundation makes the deterministic linkage from death to live birth records in order to identify birth information from all infants who died within 365 days after birth [17]”

Brasil-a. Declaração de nascido vivo: manual de instruções para preenchimento [cited 28 Dec 2022]. Available from: http://bvsms.saude.gov.br/bvs/publicacoes/declaracao_nascido_vivo_manual_4ed.pdf.

Brasil-b. Declaração de óbito: manual de instruções para preenchimento [cited 28 Dec 2022]. Available from: https://www.gov.br/saude/pt-br/centrais-de-conteudo/publicacoes/publicacoes-svs/vigilancia/declaracao-de-obito-manual-de-instrucoes-para-preenchimento.pdf/view

World Health Organization. International statistical classification of diseases and related health problems. 10th rev. Geneve: World Health Organization; 2010.

Waldvogel BC, Morais LCC, Perdigão ML, Teixeira MLP, Freitas RMV, Aranha VJ. Experiência da Fundação Seade com a aplicação da metodologia de vinculação determinística de bases de dados. Ensaio Conjuntura. 2019;1:1-25.

Nhoncanse, Geiza César e Melo, Débora Gusmão. Confiabilidade da Declaração de Nascido Vivo como fonte de informação sobre os defeitos congênitos no Município de São Carlos, São Paulo, Brasil. Ciência & Saúde Coletiva [online]. 2012, v. 17, n. 4, pp. 955-963. Disponível em: <https://doi.org/10.1590/S1413-81232012000400017>. Epub 23 Abr 2012. ISSN 1678-4561. 

Brasil. Saúde Brasil 2020/2021: anomalias congênitas prioritárias para a vigilância ao Nascimento. Available from: https://bvsms.saude.gov.br/bvs/publicacoes/saude_brasil_anomalias_congenitas_prioritarias.pdf.

• Comment 2. Line 102, The definition and classification of CDH are shown in Table 1. Why are the diagnostic ICD-10 codes, such as K44, classified as CDH? So It's important to clarify the ascertainment time of CDH. Besides, if babies are diagnosed as CDH, will all cases undergo further chromosome examination? This is important to access the trend of prevalence of CDH associated with non-chromosome abnormality.

Thank you for your question. Although Q79.0 is specific code for CDH, we chose to include codes K40, K44.0, K44.1 and K44.9 which indicate the presence of diaphragmatic hernia and may be coded in a neonate if the physician does not write the word “congenital” in the live birth or death certificate. Considering that our study was based on clinical information recorded by the physician at the time of birth and/or death of each neonate and it includes only the neonatal period, we understand that the inclusion of these codes may reduce the possibility of sample losses without overestimating the actual cases of CDH. The database used for the research contains clinical and demographic information, but it has limitations, and, among them, it is not possible to know which tests were performed or requested, including chromosomal tests, which can influence the prevalence of chromosomal and non-chromosomal anomalies associated to CDH. Thus, we understand the importance of adding a specific concern in the limitations of the study, as follows (page 17, lines 286-289): “…. thus, there is a risk of information bias since it depends on the notification of the diagnosis (including the impossibility of confirming the performance of chromosomal tests, which may have influenced the prevalence of chromosomal and non-chromosomal anomalies associated with CDH).”

• Comment 3. I would like to see the results about survival curve for newborns affected with CDH. That will give the readers more information.

Information regarding to survival curve is described in the methods (page 8, lines 137-138) and in last paragraph of results (on page 14, lines 193-197), and showed in a supplemental file 1, as follows: “Kaplan-Meier analysis showed that the median time of CDH-associated neonatal deaths was 24 hours after birth (95%CI 24 to 25 hours). For isolated CDH the median time of deaths was 26 hours (95%CI 24 to 31 hours); for CDH associated to non-chromosomal anomalies the median time to death was 24 hours (95%CI 16 to 24 hours), and for CDH associated to chromosomal anomalies, 23 hours (95%CI 7 to 48 hours) (S1 Fig.).” 

Results

• Comment 1. Line 138 of Page 8, please check the number '35.8%'. Also, please check all the data shown in Supplementary Table 1. The order of ' Other chromosomal anomalies' and 'Other trisomy' should be inter-changed.

Regarding to 35.8%, this number is correct – in 35.8% of live births with non-chromosomal anomalies associated to CDH, there was more than one group of non-chromosomal congenital anomalies registered. To better clarify this point, we included on page 9, line 149-150, the number of representing the 35.8%, as follows: “More than one anomaly group was found in 128 (35.8%) of the 358 live births with CDH associated to non-chromosomal anomalies.” Regarding the Reviewer’s concern about the interchange between 'Other chromosomal anomalies' and 'Other trisomy', the correction was made on supplementary Table 1. 

• Comment 2. Line 147-150 of page 9, could you please show 95%CI of birth prevalence of CDH in the main text.

Thank you for the suggestion. The 95%CI of CDH prevalence was inserted on page 11, lines 160-161, as follows: “CDH prevalence, during 12-year study period, was 1.67 (95%CI 1.60 to 1.77) per ten thousand live births.”

• Comment 3. Line 152 of page 9, the abbreviated 'CI' need to be defined earlier.

Thank you for the suggestion. The definition was included on page 8, lines 134-136, as follows: “Prevalence, neonatal mortality, and lethality annual trends were analyzed by Prais-Winsten Model, and their annual percent change (APC) with 95% confidence intervals (95%CI) were calculated.”

Comments from Reviewer 1

• Comment. This is an important contribution to the knowledge on CDH. The authors have presented the data clearly, pointing out the limitations of the data. My once comment relates to the use of the term incidence. Incidence is not used to report the occurrence of congenital anomalies, as the total number of conceptions cannot be measured. Instead, birth prevalence is used. This correction may be made throughout the manuscript.

Thank you for your careful review of our manuscript. We understand the question and change “incidence” for “prevalence” throughout the manuscript. 

Comments from Reviewer 2

• The authors have presented a really nice review of population based data on CDH births, mortality, and lethality in the Sao Paulo State of Brazil. The introduction and methodology of the manuscript are sound. I also thought that the results are presented clearly and succinctly for the limited data available on the topic. The discussion was thorough, but it seemed to go on for a little too long. I would recommend that the authors work to cut some of the discussion wording to make it a quicker and more streamlined read. Otherwise, I think it is worthy of publication and adds to the worldwide education about the status of CDH care in this middle-income country.

Thank you for your careful review of our manuscript and for your suggestion. We worked on the discussion and shortened it, removing the text to single-center studies.

We are grateful to the editor and to both reviewers for the suggestions to improve our manuscript. We hope that the revised manuscript fulfills the requirements of your prestigious journal. We thank you for your consideration and we are looking forward to hearing from you.

Yours sincerely,

Ana Silvia Scavacini Marinonio on behalf of all authors.

Email: anascavacini@gmail.com

---

## [Editor Report · Decision Letter 1]

31 Jan 2023

Congenital diaphragmatic hernia in a middle-income country: persistent high lethality during a 12-year period.

PONE-D-22-22879R1

Dear Dr. SCAVACINI MARINONIO,

We’re pleased to inform you that your manuscript has been judged scientifically suitable for publication and will be formally accepted for publication once it meets all outstanding technical requirements.

Kind regards,

Xiaohong Li

Academic Editor

PLOS ONE
---

## [Editor Report · Acceptance letter]

2 Feb 2023

PONE-D-22-22879R1 

Congenital diaphragmatic hernia in a middle-income country: persistent high lethality during a 12-year period. 

Dear Dr. Scavacini Marinonio:

I'm pleased to inform you that your manuscript has been deemed suitable for publication in PLOS ONE. Congratulations! Your manuscript is now with our production department. 

Kind regards, 

on behalf of

Dr. Xiaohong Li 

Academic Editor

PLOS ONE